# A Novel Polymerase Chain Reaction (PCR)-Based Method for the Rapid Identification of *Chrysodeixis includens* and *Rachiplusia nu*

**DOI:** 10.3390/insects15120969

**Published:** 2024-12-04

**Authors:** Guilherme A. Gotardi, Natália R. F. Batista, Tamylin Kaori Ishizuka, Luiz H. Marques, Mário H. Dal Pogetto, Amit Sethi, Mark L. Dahmer, Timothy Nowatzki

**Affiliations:** 1Corteva Agriscience, Rodovia SP 147 Km 71, Mogi Mirim 13801-540, SP, Brazil; natalia.batista@corteva.com (N.R.F.B.); tamylin.ishizuka@corteva.com (T.K.I.); luiz.marques@corteva.com (L.H.M.); 2Corteva Agriscience, 9330 Zionsville, Indianapolis, IN 46268, USA; mario.dal-pogetto@corteva.com; 3Corteva Agriscience, 7100 South Dr, Johnston, IA 50131, USA; amit.sethi@corteva.com (A.S.); mark.dahmer@corteva.com (M.L.D.); tim.nowatzki@corteva.com (T.N.)

**Keywords:** Plusiinae, soybean looper, sunflower looper, mtCOI, PCR, molecular identification, insect pest

## Abstract

*Chrysodeixis includens* and *Rachiplusia nu* are pests relevant to several crops due to excessive defoliation caused by their larvae leading to a significant reduction in their yield. The accurate identification of these species is crucial for effective pest management and monitoring the efficacy of engineered plants expressing insecticidal proteins and insecticides. However, visual identification is challenging due to the morphological similarities between the larvae of the two species. Molecular identification overcomes these challenges, making identification more accurate and rapid. In this research, we developed a quick and precise molecular tool for differentiating these species based on different amplicon sizes, obtaining a unique genetic profile for each species. This tool can be easily replicated in standard molecular biology laboratories, allowing early characterizations of infestations.

## 1. Introduction

Plusiinae is a well-recognized monophyletic group within the Noctuidae family. This moderately large subfamily, comprising up to 400 species, is abundant worldwide. Plusiinae larvae are semi-looping and typically polyphagous, feeding mainly on the leaves of numerous economically important crops, vegetables, and ornamental plants [1,2]. *Chrysodeixis includens* (Walker), the soybean looper, and *Rachiplusia nu* (Guenée), the sunflower looper, are the two most relevant Plusiinae pests in the Southern Cone of America and Brazil [3].

*C. includens* has been reported on 175 host plants across 39 families and is a serious pest of several crops throughout North and South America [4,5,6]. Excessive defoliation caused by larvae can reach up to 80%, leading to yield losses and increased production costs [5]. In Brazil, it is an important pest for soybean, cotton, beans, and tomato [7]. The combination of polyphagy, its high reproductive capacity with multiple generations per year, and its ability to disperse between regions and crops contributes to the occurrence of outbreaks with severe infestations during the crop seasons [6,8,9]. Thus, the fact that *Bt* soybeans can effectively control this pest explains the increasing adoption of insect-resistant soybeans in Brazil [10,11].

*Rachiplusia nu* is present in Argentina, Bolivia, Brazil, Chile, and Uruguay and has been reported to feed on sunflowers (*Helianthus annuus* L.), alfalfa (*Medicago sativa* L.), tobacco, and soybeans [12,13]. The injury caused by *R. nu* on soybeans is visually similar to that caused by *C. includens*. Although historically considered a secondary pest of soybean in Brazil with relatively low infestations, *R. nu* has recently been reported attacking *Bt* soybeans, and outbreaks can now be observed in both *Bt* and non-*Bt* soybean fields [13,14,15,16]. Altogether, the differences in control measures between *C. includens* and *R. nu* [16] make it necessary to correctly identify the species in the field, especially during the larval stage, when the damage resulted by larval feeding can result in economic losses.

Field identification of Plusiinae caterpillars is challenging due to the lack of distinctive morphological features [17]. The larvae of *C. includens* and *R. nu* are morphologically indistinguishable, leading to potential confusion [18]. This can hinder or even compromise pest control and management strategies. Accurate identification often necessitates rearing collected larvae to adulthood [17], a time-consuming process that can impact management decisions.

When morphological traits are insufficient for species discrimination, molecular identification becomes the only viable option [19]. DNA polymerase chain reaction (PCR) is a versatile technique capable of distinguishing between closely related species, including Plusiinae moths such as *Autographa nigrisigna*, *Macdunnoughia confusa*, *Thysanoplusia intermixta*, and *Autographa gamma*. This method relies on the amplification of specific DNA sequences using unique primer sets, resulting in distinct PCR product profiles [20,21,22]. Mitochondrial DNA (mtDNA) has been a cornerstone tool for identification and phylogenetic analysis since the late 1980s. A fragment of the mtDNA cytochrome c oxidase I gene (*COI*) has been established as a standard DNA barcode region for the identification of animal species [23,24]. Studies utilizing molecular tools have facilitated understanding the genetic structure and diversity of economically important pests [25,26].

The aim of this study was to develop a rapid and accurate molecular methodology to differentiate the species *C. includens* and *R. nu* based on sequence differences in the mtDNA *COI* gene. This novel method enables the swift and precise identification of numerous specimens found in the field, aiding in timely decision-making for their management.

## 2. Materials and Methods

### 2.1. Insect Populations

Two populations were utilized for the partial sequencing of the mtDNA *COI* gene in both species: *C. includens* (*n* = 5) and *R. nu* (*n* = 5). Both populations were maintained at Mogi Mirim Research Center of Corteva Agriscience, in Mogi Mirim, São Paulo, Brazil. Following the development of species-specific primers, the same populations were employed for primer validation. The primers were also validated using larvae collected in non-*Bt* soybean fields naturally infested by loopers at Mogi Mirim Research Center (SISBIO license #58435-9). Method validation was performed with field specimens collected in non-*Bt* soybeans to increase the chances of having both species co-occurring under natural infestation conditions, according to visual identification of the loopers [2,27].

### 2.2. Genomic DNA Extraction, Sequence Amplification, and Sequencing

Total genomic DNA was extracted from third-instar larvae using the DNeasy Plant Mini Kit (Qiagen, Hilden, Germany) following the manufacturer’s instructions. DNA quality and quantity were determined using a NanoDrop One spectrophotometer (ThermoFisher, Waltham, MA, USA). Samples with a 260/230 ratio ≥ 1.8 were considered to have high-quality DNA and were stored at −20 °C until their use in experiments.

The primers LCO 1490 and HCO 2198 [28] were used to amplify a fragment of mtDNA *COI* from *C. includens* and *R. nu*. The primers were synthesized by Exxtend (Paulínia, Brazil) and are listed in Table 1. High-fidelity Q5 DNA polymerase (New England Labs, Ipswich, MA, USA) was employed to amplify the gene in a reaction containing 12.5 µL of Q5 High-Fidelity 2X Master Mix, 1.25 µL (10 µM) of forward and reverse primers, and 30 ng of genomic DNA, in a total reaction volume of 25 µL. PCR amplification was performed on a Proflex thermocycler system (ThermoFisher, Waltham, MA, USA) using the following cycling conditions: 30 s of initial denaturation at 98 °C, followed by 35 cycles of 5 s of denaturation at 98 °C, 10 s of annealing at 58 °C, and 30 s of extension at 72 °C. A final extension step of 2 min at 72 °C was included. Amplicons were resolved on a 2% agarose gel stained with SYBR Safe DNA gel stain (Invitrogen, Waltham, MA, USA).

The PCR products were purified directly from the agarose gel using the Monarch DNA gel extraction kit (New England Labs, Ipswich, MA, USA), following the manufacturer’s instructions. The purified amplicon was sequenced by Omikka (São Paulo, Brazil) using the same forward and reverse primers. The sequencing reads were assembled and corrected using Chromas v2.6.6. The sequences were aligned using MEGA11, and a nucleotide BLAST search on the NCBI (https://www.ncbi.nlm.nih.gov/) (accessed on 5 February 2021) was performed to verify the accuracy of the sequences.

### 2.3. Species-Specific Primer Development

The *mtCOI* sequences of *C. includens* and *R. nu*, obtained through sequencing, were aligned to identify species-specific nucleotide sequences suitable for PCR primer design. In addition to these two species, three other species from the subfamily Plusiinae were included in the alignment: *Trichoplusia ni* (GenBank accession number: OQ564166.1), *Autographa gamma* (GenBank accession number: MF679519.1), and *Autoplusia egena* (GenBank accession number: MF679183.1) (Figure 1). A set of primers for mtCOI was designed, consisting of one reverse primer common to both species, designed at position 357–379 according to the alignment, and one forward primer specific to either *C. includens*, position 181–207 bp, or *R. nu*, position 81–102 bp. These primers were tested for their ability to amplify species-specific fragments using GoTaq G2 DNA Polymerase (Promega, Madison, WI, USA) in a reaction containing 12.5 µL of GoTaq G2 2X Master Mix, 1.25 µL (10 µM) of both species-specific forward and common reverse primers and 30 ng of genomic DNA, in a total reaction volume of 25 µL. PCR amplification was performed using the following conditions: 2 min of initial denaturation at 95 °C, followed by 35 cycles of 30 s of denaturation at 95 °C, 20 s of annealing at 56 °C, and 30 s of extension at 72 °C. The expected amplicon sizes were resolved on a 2% agarose gel stained with SYBR Safe DNA gel stain and visualized under UV light.

## 3. Results

### 3.1. mtDNA COI Partial Gene Sequencing

Following quality control and trimming, sequencing of a portion of mtDNA cytochrome c oxidase I gene (*COI*) gene for *C. includens* (GenBank accession number PQ249028-PQ249032) and *R. nu* (GenBank accession number PQ249091-PQ249095) yielded a 657-base-pair sequence for both species. The *C. includens* sequences exhibited 100% identity to previously reported sequences of the same species (GenBank accession number MT180766.1). Similarly, the *R. nu* sequences showed 100% identity with previously available sequences (GenBank accession number KC354734.1).

Individuals (*n* = 5) within each species displayed no nucleotide variation. However, between species, they exhibited 91.48% sequence similarity, with substantial and consistent differences (Figure 2). These differences were exclusively due to nucleotide substitutions, with no insertions or deletions.

### 3.2. Species-Specific Primer

Based on the sequencing data for both species, combined with the sequences of three other species belonging to the Plusiinae subfamily (*T. ni*, *A. gamma*, and *A. egena*), unique nucleotide polymorphisms were identified in *C. includens* and *R. nu*. These polymorphisms were exploited to design oligonucleotide primers. A primer set (Table 1) consisting of a conserved reverse primer and two species-specific forward primers was developed to differentiate the two species based on targeting mtCOI fragments of different sizes. The primer pair MMRC_1955 and MMRC_1964 amplified a 199 bp fragment specific to *C. includens*. The primer pair MMRC_1955 and MMRC_1988 amplified a 299 bp fragment specific to *R. nu* (Figure 3). Validation of the primers was performed using the same individuals previously identified through sequencing. All specimens were correctly identified by PCR and agarose gel electrophoresis.

### 3.3. Field Collection Validation

The feasibility of field-collected larvae identification for monitoring programs and/or rapid management decision-making was assessed through the collection of caterpillars in non-*Bt* soybean fields with natural infestation. A co-occurrence of both species was observed in the area. From the 19 larvae collected, 3 individuals (16%) were identified as *C. includens*, while the remaining 16 individuals (84%) belonged to the *R. nu* species (Figure 4).

## 4. Discussion

Traditional taxonomic methods have limitations in differentiating morphologically similar species, often requiring significant time and specialized knowledge [28]. Molecular techniques offer a promising alternative for addressing these challenges. The present study successfully developed a molecular tool for the rapid and accurate identification of *C. includens* and *R. nu*, two morphologically similar lepidopteran species.

The mtCOI gene sequencing revealed a high degree of genetic divergence between the two species, despite their morphological similarity. Based on these differences, a set of primers was developed to differentiate *C. includens* and *R. nu* through PCR amplification.

DNA barcoding has been proven to be a powerful method for rapid species identification [29]. The primers described in this study are highly specific and sensitive, enabling the accurate identification of both species. Our results with field-collected larvae for validating the method indicated that both *C. includens* and *R. nu* were present in the samples with a higher prevalence of *R. nu*. Our findings differ from a previous study reporting the predominance of *C. includens* over *R. nu* in non-*Bt* soybean based on the morphological identification of loopers at the larval stage [10]. This contrast highlights the importance of a reliable identification method that can detect even low frequencies of individuals and eliminate misidentifications associated with traditional taxonomic methods to identify caterpillars of these species and of the Plusiinae subfamily [30,31]. This study provides a valuable tool for field-collected larvae with practical utility through a rapid and reliable alternative to traditional taxonomic identification of species and may reduce the complexity of monitoring and improve pest management of *C. includens* and *R. nu*.

## Figures and Tables

**Figure 1 insects-15-00969-f001:**
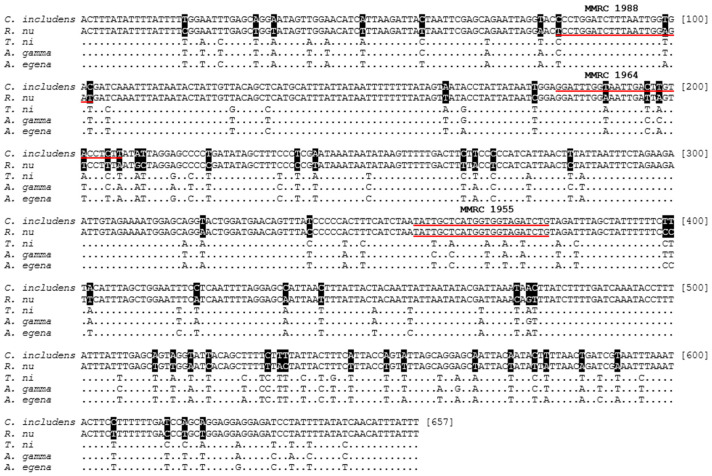
Alignment of partial mtCOI gene sequences for *C. includens*, *R. nu*, *Trichoplusia ni*, *A. gamma*, and *A. egena*. The black dots indicate the nucleotide positions conserved across all the species. The nucleotides in white text over black backgrounds highlight differences between *C. includens* and *R. nu*. The sequences underlined in red indicate the position of the primers.

**Figure 2 insects-15-00969-f002:**

Multiple sequence alignment of the mitochondrial cytochrome c oxidase subunit I (mtCOI) gene region for *R. nu* and *C. includens*, highlighting nucleotide variability between the species. The reference sequences used were KC354734.1 for *R. nu* and MT180766.1 for *C. includens*. Identical nucleotides are displayed in white text over a black background, while mismatched nucleotides are shown in plain text.

**Figure 3 insects-15-00969-f003:**
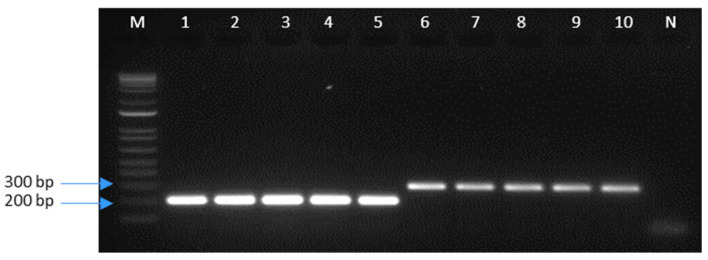
SYBR Safe-stained 2% agarose gel electrophoresis of mtCOI PCR products. Lanes 1–5 show a 199 bp amplicon confirming *C. includens*. Lanes 6–10 show a 299 bp amplicon indicating *R. nu*. Lane M contains a 1 kb DNA ladder for size estimation, and lane N is a negative control.

**Figure 4 insects-15-00969-f004:**
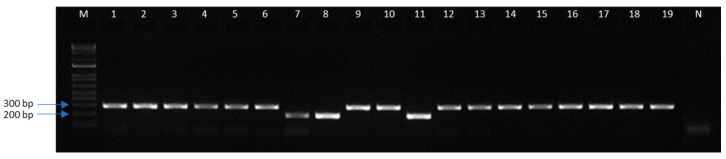
Agarose gel electrophoresis of mtCOI gene amplicons for validating method with field-collected insect samples. M: 1 kb DNA ladder; Lanes 1–6, 9, 10, and 12–19: 299 bp amplicon for *R. nu*; Lanes 7, 8, and 11: 199 bp amplicon for *C. includens*.

**Table 1 insects-15-00969-t001:** Nucleotide sequences of primers used for PCR amplification.

Primer Name	Primer Sequence (5′ > 3′)	Purpose
LCO 1490	GGTCAACAAATCATAAAGATATTGG	mtDNA COI amplification and sequencing
HCO 2198	TAAACTTCAGGGTGACCAAAAAATCA	mtDNA COI amplification and sequencing
MMRC_1955	CAGATCTACCACCATGAGCAATA	Species Identification—Common reverse
MMRC_1964	GGATTTGGTAATTGACTTGTACCTCTT	Species Identification—*C. includens*-specific forward
MMRC_1988	TCCTGGATCTTTAATTGGAGAT	Species Identification—*R. nu*-specific forward

## Data Availability

The original contributions presented in this study are included in this article; further inquiries can be directed to the corresponding author.

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
