# Peer review of "A Novel Polymerase Chain Reaction (PCR)-Based Method for the Rapid Identification of Chrysodeixis includens and Rachiplusia nu"

_insects, 2024, doi:10.3390/insects15120969_

Round 1
Reviewer 1 Report
Comments and Suggestions for Authors
This research addresses an interesting issue concerning the accurate and rapid identification of two lepidopteran pest species.While the study presents valuable findings, I have several inquiries as follows:
Do Chrysodeixis includens and Rachiplusia nu exhibit differences in their life cycle timings or in the control methods which necessitate their discrimination during the larval stages?
Why did you choose to collect larvae exclusively from non-Bt fields? Would it not be beneficial to include larvae from both non-Bt and Bt fields in your study?
Do you think it is important to use the other species of the genus in the alignment in order to make the primer specific to the target species? How did you confirm that no cross-reactivity occurred with non-target species during the primer validation process?
Author Response
1. Summary |
|
|
Thank you very much for taking the time to review this manuscript. We truly appreciate the points raised. Please find the detailed responses below and the corresponding revisions/corrections highlighted in the re-submitted files.
|
||
2. Questions for General Evaluation |
Reviewer’s Evaluation |
Response and Revisions |
Does the introduction provide sufficient background and include all relevant references? |
Yes |
|
Are all the cited references relevant to the research? |
Yes |
|
Is the research design appropriate? |
Can be improved |
The responses to your comments are presented in a point-by-point format below. |
Are the methods adequately described? |
Yes |
|
Are the results clearly presented? |
Yes |
|
Are the conclusions supported by the results? |
Yes |
|
3. Point-by-point response to Comments and Suggestions for Authors
|
||
Comments 1: Do Chrysodeixis includens and Rachiplusia nu exhibit differences in their life cycle timings or in the control methods which necessitate their discrimination during the larval stages?
|
||
Response 1: Thank you for pointing this out. Yes, there are significant differences in the control of both species. In Brazil, C. includens is primarily managed through the use of Bt cultivars, as the species is highly susceptible to Bt toxins. Conversely, R. nu does not exhibit susceptibility to Bt proteins and can feed on both Bt and non-Bt cultivars. Additionally, the two species exhibit different susceptibilities to commercial insecticides in Brazil, including Bt-based products. Considering these differences, monitoring both species is crucial to understand their distribution following the adoption of Bt technology and to monitor the development of resistance to Bt and other insecticides in these pests. To clarify this point, we have reformulated the sentence on line 53-55 of the second paragraph of the introduction. Furthermore, we have added a more detailed explanation of the differences in control strategies for both species in the third paragraph of the introduction, lines 62-64. Comments 2: Why did you choose to collect larvae exclusively from non-Bt fields? Would it not be beneficial to include larvae from both non-Bt and Bt fields in your study?
Response 2: Thank you for your question. It would be very unlikely to find Chrysodeixis includens in Bt soy as mentioned above. We were interested in validating the PCR results with field specimens in addition to the validation with laboratory colonies, thus we collected specimens from non-Bt soybean to be certain we would have both species. To clarify this point, we added this information in Material and Methods, “2.1 Insects Populations” on line 96-99.
Comments 3: Do you think it is important to use the other species of the genus in the alignment in order to make the primer specific to the target species? How did you confirm that no cross-reactivity occurred with non-target species during the primer validation process?
Response 3: Thank you for bringing this to our attention. We agree that the inclusion of additional species in the alignment for primer design is crucial. While this was indeed done during primer design, we acknowledge that this information was omitted from the text. To address this, we have added a figure (Figure 1) on page 4 that presents the alignment of sequences from all species used. Additionally, we have included a more detailed description of the primer design methodology in section 2.3, 'Species-Specific Primer Development', on page 3, lines 128-133.
|
||
4. Response to Comments on the Quality of English Language |
||
Point 1: |
||
Response 1:
|
||
5. Additional clarifications |
||
The main modifications, including your suggestions and those of the other editors, are highlighted in red for easier tracking. |
Reviewer 2 Report
Comments and Suggestions for Authors
Given that we are unable to differentiate between Chrysodeixis and Rachiplusia nu morphologically, this is probably a rather intriguing and eventually significant attempt of investigation.
However, I would advise against publishing it as a main piece because it merits much more effort and consideration. Instead, I would suggest publishing it as a brief communication. The only result, after all, is the makeup of the primers that we may utilize to aid in the larvae's (in the lab) differentiation.
The title should state that it is merely a PCR experiment and that a set of primers has been reported.
For short communication, I would recommend to discuss the following:
- Given that Chrysodeixis has distinct genetic structures and molecular variability, how does it aid in the differentiation of individuals within a species (see Palma et al., 2015)?
- How can it be certain that the species taken from the field are Chrysodeixis includens (Walker) or Rachiplusia nu (Guene ́ e) when several species of Lepidoptera are known to attack soybeans and present larvae that make species identification challenging?
- How can these primers help differentiate Chrin and Raclu from the other most prevalent Plusiinae species?
- To see if it could function without the genomic DNA extraction procedure, the PCR results utilizing insects crushed in PCR buffer should be displayed.
- Results of PCR utilizing the two species' complete developmental profiles should be displayed separately; for example, compare the two plusinaee's eggs, larvae, nymphs, and adults using PCR and these primers.
- The location of the primers and what is species-specific should be revealed by the sequence alignment. The length of the anticipated PCR products should also be indicated. It is somewhat strange to use DNA as a sample and generate two PCR products that differ by 100 bps when there is no gap between the primers. When Rnu and Cin are compared in Figure 1, the mitochondrial COX1 matches exactly. According to this alignment, the final PCR products ought to be precisely the same size, and the only way to determine the distinctions between Cin and Rnu would be by sequencing.
- Please supply the molecular sequences that were obtained for the RNU and CIN PCR products.
- To make generalizations regarding the effectiveness of the "new primers" approach, please provide the PCR results and molecular sequences for RNA and CIN derived from various crops.
Round 2
Reviewer 2 Report
Comments and Suggestions for Authors
A phylogenetic analysis of the COI sequences from Chrysodeixis includens and Rachiplusia nu would have been a nice addition